# Effects of Simulated Herbivory on the Vegetative Reproduction and Compensatory Growth of *Hordeum brevisubulatum* at Different Ontogenic Stages

**DOI:** 10.3390/ijerph16091663

**Published:** 2019-05-13

**Authors:** Jihong Yuan, Ping Wang, Yunfei Yang

**Affiliations:** 1Key Laboratory of Vegetation Ecology, Ministry of Education, Institute of Grassland Science, Northeast Normal University, Changchun 130024, China; yuanjh040@nenu.edu.cn; 2State Environmental Protection Key Laboratory of Wetland Ecology and Vegetation Restoration, Northeast Normal University, Changchun 130117, China; wangp744@nenu.edu.cn

**Keywords:** tussock grass, potential vegetative propagation, compensation index, clipping, phenological period, ontogeny

## Abstract

The response of plant vegetative reproduction and compensatory growth to herbivory has been widely discussed in biological and ecological research. Most previous research has supported the idea that both vegetative reproduction and compensatory growth are affected by their ontogenic stage. However, in many studies, the effects of foraging at different ontogenic stages was often confounded with the effects of foraging at different phenological periods for perennials. Our experiment was conducted in a natural meadow with a perennial grass, *Hordeum brevisubulatum,* and four ontogenic stages were chosen as our experimental objects. Three different clipping intensities during three phenological periods were implemented to explore the effects of simulating animal foraging on vegetative reproduction and compensatory plant growth. The results indicated that there were significant effects of ontogenic stage, phenological period, and clipping intensity on vegetative reproduction and compensatory growth. Moderate clipping intensities significantly increased the number of vegetative tillers, the total number of juvenile tillers and buds, and the aboveground biomass at early phenological periods for individuals at early ontogenic stages. Our results suggested that moderate clipping intensities could induce only an over-compensation response in perennial grasses at both the early ontogenic stage and phenological period, and the ability of compensatory growth gradually decreased with the progression of the ontogenic stage. This is of great significance to the primary production of grasslands subjected to herbivory.

## 1. Introduction

Plants foraged partially or entirely by herbivores are common in grassland ecosystems. Normally, there are three responses of plants to herbivory, which are under-compensation, complete compensation, and over-compensation, as the results of the co-evolution of plant–herbivore interactions [1]. Compensatory plant growth is defined as the ability of plants to offset the adverse effects of tissue damage, restore organic functionality, and maintain normal growth after herbivore foraging [2]. The changes in the activity of lateral meristems after breaking of apical dominance [3], the leaf photosynthetic rate [4], the relative growth rate [5], and the redistribution of carbohydrates [6,7,8] have been considered to be the mechanisms of plants to compensate for lost tissue. The breaking of apical dominance after herbivore foraging could initiate the activity of the lateral meristems. For perennial grasses, herbivore foraging would promote the vegetative reproduction of plants and then cause plants to show over-compensation growth [9,10]. Many studies have shown that the response of vegetative reproductive capacity to herbivore foraging plays a decisive role in the compensatory growth of grassland plants, especially for perennial grasses in grassland ecosystems [9,10,11,12].

The capacity for vegetative reproduction and compensatory growth due to herbivore damage is often influenced by plant species [13,14,15], plant life forms [16,17], plant growth status when being foraged [18,19], resource availability [20,21], foraging time [6,22], and intensity [18,23]. The type of compensatory growth may vary when foraging occurs at different ontogenic development stages of plants because their organ structures, storage capacities, physiological regulations, and recovery periods after foraging are different at different ontogenic stages [24]. Many studies have shown that plants at early ontogenic stages are less vulnerable to foraging and that the compensation capability is greater than that for those plants foraged at later ontogenic stages [6,25]. However, other studies have shown that the negative effects on plant growth of herbivore foraging were higher at early ontogenic stages [17]. The conclusions about the effect of ontogenic stage in plants is inconsistent.

The response process of plant compensatory growth to clipping at different ontogenic stages is generally complex and non-linear [26]. Ontogeny is the developmental history of an individual organism, including the entire sequence of events involved in the development of an individual organism. Most previous studies on the response of compensatory growth to ontogenic stage were based on a certain growing time (phenological period), which regarded clipping at the early phenological stage as the early ontogenic stage and the later phenological stage as the later ontogenic stage [6,16,17,18,19,26,27,28,29]. For annuals, this is true, i.e., clipping at different ontogenic stages is consistent with the phenological period. However, for perennials, the early ontogenic stages generally refer to the first year or first several years of the entire life history of the plant, and the later ontogenic stage refers to the last one or the last several years before plant death. As the ontogenic stage is inconsistent with the phenological period, it is necessary to distinguish between the ontogenic stage and the clipping time when studying the compensatory growth response to clipping.

*Hordeum brevisubulatum* is a perennial grass species with short rhizomes; it grows in rather dense tufts and is considered to be a quality forage grass with high feeding value due to its high yield, good palatability, and strong tolerance to salt. This grass species is widely distributed in the grasslands of northeastern China [30]. To study the difference in plant compensatory growth at different ontogenic stages, we chose individuals of *H. brevisubulatum* at different ontogenic stages. To explore the effects of clipping intensities and clipping times on the vegetative reproduction and compensatory growth of plants, this study was conducted with different clipping intensities when the plants were clipped at the jointing, booting, and heading stages. The objective of this study was to explore the effects of clipping on the vegetative reproduction and compensatory growth of plants at different ontogenic stages. Specifically, we mainly addressed the following questions: (1) How does the vegetative reproduction of plants at different ontogenic stages vary among different clipping treatments, and does clipping promote vegetative reproduction? (2) How does the compensatory growth of plants at different ontogenic stages vary among different clipping treatments, and would there be over-compensation growth after clipping?

## 2. Materials and Methods

### 2.1. Experimental Site

The experiment was conducted in a natural meadow located at the Ecosystem Field Station of the Institute of Grassland Science in the southern Songnen Grassland in Changling County, Jilin province (45°45′ N, 123°45′ E), which is in a semi-arid area with a continental monsoonal climate. The annual average temperature is 4.9 °C, and the annual rainfall and evaporation are 470.6 and 1668 mm, respectively. The frost-free period is 150 days [31]. The soil type is alkaline.

Due to unreasonable historical utilization, this entire natural meadow had been degraded into many patches dominated by *Leymus chinensis*, a dominant species in this meadow. Natural restoration succession is now ongoing for the spaces between these patches, where halophyte communities were established. Therefore, the meadow is characterized by a mosaic distribution of an *L. chinensis* community and by halophyte communities. *Hordeum brevisubulatum*, one of the dominant species in the halophyte communities, was chosen as the experimental plant. The total coverage of the *H. brevisubulatum* community is less than 50%, and the associated species of this community consist of several halophytes and salt-tolerant species, including *Polygonum sibiricum*, *Taraxacum sinicum*, *Puccinellia tenuiflora,* and *Chloris virgata*.

### 2.2. Plant Species

*H. brevisubulatum* is a perennial grass species with short rhizomes; it grows in rather dense tufts and is considered to be a quality forage grass with high feeding value due to its high yield, good palatability, and strong tolerance to salt. This grass species is widely distributed in the grasslands of northeastern China [30]. *H. brevisubulatum* normally turns green in early April and flowers in early June, and seed maturation occurs in late June. The renovation and re-establishment of natural populations of *H. brevisubulatum* depend on both sexual and asexual reproduction. Seeds that fall onto the soil germinate and grow into seedlings in the same year and for several later years when the rainfall is sufficient [32]. The tuft size of this plant increases with increasing years of growth, and many different tuft sizes are randomly distributed in the halophyte communities due to the seed germination taking place in different years. According to our preliminary investigation, the maximum tuft diameter (D) of *H. brevisubulatum* is up to 50 cm in this meadow. As the coverage of the *H. brevisubulatum* community is quite low (< 50%), the individuals of *H. brevisubulatum* grow in a relatively isolated manner, so the size of tufts can represent the ontogenic stage.

### 2.3. Experimental Design

Three factors were considered in this experiment, which were ontogenic stage, phenological stage, and clipping intensity. The tuft diameters of the plants were divided into four grades to represent different ontogenic stages: D ≤ 10 cm (grade 1), 10 < D ≤ 20 cm (grade 2), 20 < D ≤ 25 cm (grade 3), and D > 25 cm (grade 4). Clipping was conducted at the jointing stage (May 10), booting stage (May 18), and heading stage (May 25) in 2017. Three clipping intensities were performed for each ontogenic stage, including 0% (non-clipped treatment), 50%, and 100%, as shown in Figure 1. The heights of the clipping stubble were all maintained at 5 cm. There were 5 repetitions of each treatment, and a total of 180 plants in the natural community were selected and measured. To avoid the influence of the environment on the growth of *H. brevisubulatum*, all the plants were selected from within a nearly square area of approximately 100 m^2^.

### 2.4. Harvest and Measurement

All plants of *H. brevisubulatum* were harvested on July 10, 2017, when the seeds were completely mature. The tuft diameters of *H. brevisubulatum* were measured from three different directions; then, the whole plant was dug out. The aboveground tillers were divided into reproductive tillers, vegetative tillers, and juvenile tillers. The juvenile tillers were defined as small vegetative tillers without jointing. The number of reproductive tillers, vegetative tillers, juvenile tillers, and buds at the tiller nodes were counted. All kinds of tillers were dried for 48 h at 80 °C and then weighed to obtain the biomass. The tuft diameters (the average value of tuft diameter from three different directions), total number of juvenile tillers and buds (the sum number of the juvenile tillers and buds), and the aboveground biomass (the sum of the biomass of reproductive tillers, vegetative tillers, and juvenile tillers) were calculated.

### 2.5. Data Analysis

To eliminate the effect of variations in diameter on all indexes within one diameter grade, all data in each grade were standardized based on the mean diameter calculated for every grade. The standardization method was as follows:(1)$$X_{si}=X_{i\;}\cdot \;D_{si}/D_{i}$$
where *X_si_* and *X_i_* represent the value of an index after and before standardization, respectively (*i* represents the ontogenic stage of tufts, the values of *i* were 1, 2, 3, and 4). D*_si_* is the mean tuft diameter of each grade, and these were 8, 17, 23, and 29 cm for grades 1, 2, 3, and 4, respectively. D*_i_* is the actual value of each individual tuft diameter.

The compensatory growth ability of each treatment is calculated as [9]
(2)$$CI=\;B_{T}/B_{ck}$$
where *CI* is the compensation index, and *B_T_* and *B_ck_* are the aboveground biomass of plants under clipping treatments and under non-clipping treatments, respectively. A value of *CI* greater than 1 means over-compensation growth, values equal to 1 mean complete compensation growth, and values less than 1 mean under-compensation growth.

Data were transformed when necessary to achieve normality and homogeneity of variances. Multivariate analyses of variance were used to analyze the main and interactive effects of ontogenic stage, phenological period, and clipping intensity on the number and biomass of the vegetative tillers, the total number of juvenile tillers and buds, the aboveground biomass, and the compensation index of the plants. The Least Significant Difference (LSD) method of one-way ANOVA was used to make multiple comparisons to the above indexes between different clipping intensities. The differences between the compensation index and 1 were compared with the use of a single-sample *t-*test, and the difference in the compensation index between two clipping intensities was compared by an independent-sample *t-*test. All data used have been standardized, and the statistical analysis was performed using SPSS 20.0 statistical software (SPSS Inc., Chicago, IL, USA).

## 3. Results

The ANOVA test results showed that there were significant effects from ontogenic stage, phenological period, clipping intensity, and their interactions on the growth of *H. brevisubulatum*, especially for the aboveground biomass and for the compensation index, as shown in Table 1.

### 3.1. Vegetative Tillers

For grade 1 tufts, clipping treatments at the jointing and booting stages increased the number of vegetative tillers, but only a 50% clipping intensity showed a significant level, as shown in Figure 2A. For grade 2 tufts, different clipping intensities at the jointing and booting periods did not significantly affect the number of vegetative tillers. Compared to the non-clipped treatment, the 100% clipping level at the heading period significantly decreased the number of vegetative tillers, as shown in Figure 2B. For grade 3 tufts, clipping at the jointing period increased the number of vegetative tillers and reached a significant level for 100% of the clipping treatments. There was no significant difference in the number of vegetative tillers among different clipping intensities for the booting and heading periods, as shown in Figure 2C. For grade 4 tufts, different clipping intensities at the jointing and heading periods did not significantly affect the number of vegetative tillers. The number of vegetative tillers under the 100% clipping intensity at the booting period was greater than that for the 50% clipping intensity, and there was no significant difference between both clipping intensities and the non-clipped treatment, as shown in Figure 2D.

The effect of clipping intensities at different clipping times on the biomass of vegetative tillers was similar to the number of vegetative tillers. The biomass of vegetative tillers for grade 1 tufts under the 50% clipping treatment at the jointing period and for grade 3 tufts under the 100% clipping treatment at the jointing period significantly increased compared to those under the non-clipped treatment, as shown in Figure 3A,C. However, the 100% clipping level at the heading period for grade 2 tufts significantly decreased the biomass of the vegetative tillers, as shown in Figure 3B.

### 3.2. Total Number of Juvenile Tillers and Buds

For grade 1 tuft diameters of *H. brevisubulatum*, compared to the non-clipped treatment, clipping increased the total number of juvenile tillers and buds and reached a significant level under 50% clipping treatments at the booting and heading periods, as shown in Figure 4A.

However, for grade 2 tufts, different clipping treatments decreased the total number of juvenile tillers and buds and showed a significant decrease under both the 50% and 100% clipping levels at the jointing period and for the 50% clipping level at the booting period, as shown in Figure 4B.

For grade 3 tufts, compared to the non-clipped treatment, different clipping intensities did not show significant effects on the total numbers at the jointing and heading periods. The total number of juvenile tillers and buds showed a significant decrease under the 50% clipping treatments at the booting period when compared with that under the non-clipped treatment, as shown in Figure 4C.

For grade 4 tufts, there was no significant difference in the total number of juvenile tillers and buds among different clipping intensities at the jointing and heading periods. The number of juvenile tillers and buds under the 100% clipping intensity at the booting period was higher than that under the 50% clipping intensity, but the number of juvenile tillers and buds under both the 50% and 100% clipping levels were similar with that under the non-clipped treatment, as shown in Figure 4D.

### 3.3. Aboveground Biomass and Compensation Index

For grade 1 tufts, compared to the non-clipped treatment, clipping at the jointing period increased the aboveground biomass and reached a significant level under the 50% clipping level, indicating that moderate clipping intensities promoted over-compensation growth (*t* = 10.675, *p* < 0.001) and that severe clipping intensities promoted complete compensation growth (*t* = 1.514, *p* = 0.205). There was no significant difference for the aboveground biomass among the different clipping intensities at the booting and heading periods, indicating that different clipping intensities at the booting (50%: *t* = 1.872, *p* = 0.120; 100%: *t = –*2.514, *p* = 0.066) and at the heading periods (50%: *t* = 2.547, *p* = 0.064; 100%: *t = –*0.816, *p* = 0.460) could induce complete compensation growth for the grasses, as shown in Figure 5A.

For the larger tufts of *H. brevisubulatum*, the influence on the aboveground biomass of clipping at the jointing period was lower than that of clipping at the other phenological periods. For grade 2 tufts, the aboveground biomass under a 50% clipping treatment at the jointing period was higher than that of the non-clipping treatment and induced an over-compensation growth response (*t* = 4.442, *p* = 0.007). However, clipping under the 100% clipping level showed under-compensation growth (*t = –*3.838, *p* = 0.018), as shown in Figure 5B. Clipping at the jointing period significantly decreased the aboveground biomass under both the 50% and 100% clipping treatments for grade 3 (50%: *t = –*10.760, *p* < 0.001; 100%: *t = –*5.425, *p* < 0.001) tufts and for grade 4 (50%: *t = –*2.782, *p* = 0.050; 100%: *t = –*7.310, *p* = 0.002) tufts, inducing under-compensation growth, as shown in Figure 5C,D. Different clipping intensities at the booting and heading periods for grade 2 (Booting period, 50%: *t = –*8.190, *p* = 0.001; 100%: *t = –*8.558, *p* < 0.001. Heading period, 50%: *t = –*13.766, *p* < 0.001; 100%: *t = –*8.219, *p* < 0.001), grade 3 (Booting period, 50%: *t = –*20.449, *p* < 0.001; 100%: *t = –*57.026, *p* < 0.001. Heading period, 50%: *t = –*33.897, *p* < 0.001; 100%: *t = –*19.739, *p* < 0.001) and grade 4 (Booting period, 50%: *t = –*3.089, *p* = 0.037; 100%: *t = –*17.939, *p* < 0.001. Heading period, 50%: *t = –*11.787, *p* < 0.001; 100%: *t = –*17.947, *p* < 0.001) tufts induced under-compensation growth, as shown in Figure 5B–D.

There was a significant effect of clipping intensity, clipping time, and ontogenic stage on the compensation index of *H. brevisubulatum*. The compensation index under the 50% clipping level at different phenological periods was higher than for the 100% clipping level. The compensation index was significantly highest when clipping was at the jointing period or for grade 1 tufts. Furthermore, the ability for compensatory growth after clipping was higher when clipping at the jointing period than for other later phenological periods, and the ability for compensatory growth decreased with the development ontogenic stages, as shown in Figure 6.

## 4. Discussion

### 4.1. The Effects of Clipping on the Vegetative Reproduction of Plants

For clonal plants, the number of vegetative tillers can indicate the ability of vegetative propagation [33] and can be affected by clipping treatment [3,13,34,35,36]. In our experiment, the clipping intensities, phenological periods, and ontogenic stages affected the growth of vegetative tillers, and the biomass or the number of vegetative tillers was significantly higher under the 50% clipping treatment at the jointing and booting periods for grade 1 tufts, as shown in Figure 2 and Figure 3. The results indicated that the growth of vegetative tillers tended to increase significantly when moderate clipping was performed at the early phenological period for individuals at early ontogenic stages. Our results were consistent with those of previous studies indicating that moderate clipping intensities promoted plant growth [8,37,38,39].

Defoliation by herbivores can break apical dominance, stimulating the activity of lateral meristems to develop more ramets and enlarge the photosynthetic area of plants to compensate for damage [3]. Many studies on trees and shrubs have suggested that clipping can promote the development of lateral branches and alter the canopy structure [36,40]. Studies on herbaceous plants have found that clipping promotes the growth of vegetative tillers and increases the ramet density [3,35]. However, some studies have pointed out that clipping treatments can also inhibit the growth of vegetative tillers [13]. These contradictory results may be related to plant species [41,42], the growth status of plants [14], resource availability levels [8,43], clipping intensity [8,44], and clipping time [22,45]. In our study, the number of vegetative tillers after clipping either increased or remained unchanged compared to the non-clipped treatments. Additionally, the number and biomass of vegetative tillers increased more when clipping took place at early phenological periods rather than at later phenological periods, as shown in Figure 2 and Figure 3. The results indicated that clipping could promote the germination of vegetative tillers and increase the number of vegetative tillers to the same level or even higher than that for the non-clipped treatments.

The proportions of the number of juvenile tillers and buds in the total number of vegetative tillers, juvenile tillers, and buds (the total vegetative reproduction ability) ranged from 31.82% to 87.67%, and the proportions were more than 50% in most cases (calculated from the data of the number of vegetative tillers and the total number of juvenile tillers and buds in Figure 2 and Figure 4, respectively). The juvenile tillers and buds were the main components of the potential parts of vegetative reproduction. Clipping treatments affected the total number of juvenile tillers and buds. For tufts at an early ontogenic stage, the total number of juvenile tillers and buds increased or remained unchanged after clipping. For tufts at later ontogenic stages, the different clipping intensities decreased or did not affect the total number of juvenile tillers and buds, as shown in Figure 4. The results suggested that the total number of juvenile tillers and buds significantly increased for individuals at an early ontogenic stage with moderate clipping intensity at the booting and heading stages. The significant increase in the total number of juvenile tillers and buds under some treatments was important to the compensation growth of plants after clipping and to the improvement in grassland productivity [11]. This is mainly because the juvenile tillers and buds were generally not easily foraged by herbivores and could rapidly develop into tillers when conditions were suitable. Clipping at different phenological periods had no significant effect on the total number of juvenile tillers and buds, as shown in Table 1. The main reason is that grazing may facilitate the outgrowth of belowground buds into aboveground shoots, leading to a decrease in bud density for grazed sites [12]. The total number of juvenile tillers and buds could not increase cumulatively due to the output of juvenile tillers and buds into aboveground tillers.

In summary, our results found that clipping treatments could promote the vegetative reproduction of *H. brevisubulatum*, but this improved effect was related to the clipping intensities, phenological periods, and ontogenic stages of this plant. The increase in the number of, or biomass of, vegetative tillers when clipping that took place at earlier phenological periods was higher than that at later phenological periods. There was no significant difference in the response of the total number of juvenile tillers and buds to clipping at different phenological periods. The negative effect of clipping on the number of, or the biomass of, the vegetative tillers and the total number of juvenile tillers and buds for individuals at earlier ontogenic stages was lower than that at later ontogenic stages. The results indicated that moderate clipping intensities significantly increased the number of vegetative tillers and the total number of juvenile tillers and buds at early phenological periods for individuals at early ontogenic stages due to the breaking of apical dominance. In grasslands dominated by perennial grasses, the maintenance and regeneration of the aboveground parts of the plant community depend mainly on vegetative reproduction, and the role of seed reproduction is very small [11,46,47]. In our study, the improvement in the vegetative reproductive capacity of plants promoted the occurrence of over-compensation after clipping. Therefore, in research on the compensatory growth of perennial grasses, the measurement of the compensatory growth capacity of plants not only should consider the changes in plant biomass or production but also needs to consider the changes in the vegetative reproductive capacity of plants.

### 4.2. The Effects of Clipping on the Compensation Growth of Plants

Our results showed that the clipping intensities, phenological periods, and ontogenic stages affected the aboveground biomass and the compensatory growth of *H. brevisubulatum*. The plants may show over-compensation, complete compensation, or under-compensation under different clipping intensities, and moderate clipping levels at the jointing period for individuals at early ontogenic stages could promote plant growth, which then would more easily induce an over-compensation growth response, as shown in Figure 5. It is generally agreed that an over-compensation growth response requires specific conditions, suggesting that only moderate levels of defoliation could induce over-compensation, while light or severe foraging intensities inhibit plant growth [7,8]. In our study, over-compensation appeared only under moderate clipping intensities when clipping took place at early phenological periods for individuals at early ontogenic stages. Many previous studies have suggested that the main factor for limiting the occurrence of plant over-compensation in grassland ecosystems was the low resource availability (such as water and nutrients) in natural environments [20,21]. The intrinsic characteristics of plants, such as the characteristics of ontogenic stage were often ignored. The individuals from early ontogenic stages were more likely to exhibit over-compensatory growth in our study.

Plants may show different compensatory growth patterns under different clipping intensities [6,14,42]. In our study, moderate clipping intensities could significantly increase, decrease, or have no significant effect on the aboveground biomass for individuals at different ontogenic stages when clipped at the jointing and booting periods. We suggest that the effects of clipping intensity on the compensation growth of plants depends on the clipping time and on the ontogenic stage [15,48]. The negative effects of clipping at early phenological periods was lower than that at later phenological periods; when the clipping happened at an early phenological period, the time for recovery growth after clipping was longer, resulting in a stronger ability for compensatory growth. This result is supported by many other studies [6,16,49]. The results indicated that moderate clipping intensities at early phenological periods could promote over-compensation growth in plants, which was conducive to guide the maintenance of grassland productivity and the management of grassland ecosystems.

Smaller tufts of *H. brevisubulatum* showed over-compensation or complete compensation, while the larger tufts showed complete compensation or under-compensation, as shown in Figure 5. This result indicated that the ability for compensatory growth was higher for plants at early ontogenic stages than for those at later ontogenic stages. The main reason was that young individuals at early ontogenic stages were more vigorous and more resistant to interference, while older plants at later ontogenic stages had relatively weak growing ability [50].

For perennials, the response of plants at different ontogenic stages to foraging by herbivores is generally complex. Boege and Marquis (2005) indicated that the response of plants to defoliation was not a simple linear relationship with the change in ontogenic stage. The ability for compensatory growth may increase at first and then decrease with the development of ontogenetic stage [26]. This was related to the clipping time at different ontogenic stages, as shown in Figure 7. As in the study of removing cotyledons, when the time of cotyledon removal was earlier, the negative effects on plant biomass were greater. The main reason for this result was that cotyledons were the main photosynthetic organs and were important for plant growth at early stages [22]. In the clipping experiment of monocarpic perennial herbs, Tito et al. (2016) indicated that the ability for compensatory growth showed no significant change at different clipping times during the juvenile non-reproductive vegetative growth stage. During the pre-reproductive and reproductive stages, the ability for compensatory growth was higher when clipping at early developmental times than when clipping at later developmental times [18].

In our study, the clipping treatments occurred at the pre-reproductive period of the early or later ontogenic stages. The negative effects of clipping on compensatory growth for individuals at the reproductive period were higher than those for individuals at the pre-reproductive period (unpublished data). For individuals at the early ontogenic stage, our results showed that the ability for compensatory growth decreased with the developmental time. Unlike monocarpic perennial herbs, *H. brevisubulatum* shows sexual reproduction every year after the year of sowing. Therefore, for individuals at later ontogenic stages, we also find that the ability for compensatory growth decreases with the developmental time, as shown in Figure 7. The ontogenic stage could affect the ability for compensatory growth after clipping with a quite complex process for perennial plants, which was ignored in most previous research, as the ontogenic stages and phenological periods were confusing. Our results indicated that the ability for compensatory growth was highest when clipping occurred at early phenological periods for individuals at early ontogenic stages. The response rule of plant compensatory growth to clipping during different development times at different ontogenic stages plays an important role in guiding the restoration of degraded grasslands and in the management of grassland production. Therefore, to maintain the productivity and to ensure sustainable utilization of grassland ecosystems, moderate intensities of grazing and clipping, irrigation and fertilization should be applied to grassland ecosystems. Additionally, we need to consider the grazing or clipping times and the characteristics of the ontogenetic stages of the plants. Plants at early ontogenetic stages have strong vitality and have a strong ability for compensatory growth and to resist pressure from environmental changes. We can regenerate grassland plants by artificial sowing to increase the number of individual plants at earlier ontogenetic stages in the management of grassland ecosystems.

## 5. Conclusions

Our study demonstrated that there were significant effects of ontogenic stage, phenological period, and clipping intensity on the vegetative reproduction and compensatory growth of *H. brevisubulatum*. These plants may show under-compensation, complete compensation, and over-compensation after clipping for different clipping intensities during three phenological periods at four ontogenic stages. Moderate clipping intensity could only induce an over-compensation response for perennial grasses at both the early ontogenic stage and the early phenological period, and the ability for a compensatory growth response to clipping gradually decreased with the plant progression to a later ontogenic stage. The results regarding plant vegetative reproduction and compensatory growth to clipping during different development times and at different ontogenic stages play an important role in guiding the restoration of degraded grasslands and in the management of grassland production.

## Figures and Tables

**Figure 1 ijerph-16-01663-f001:**
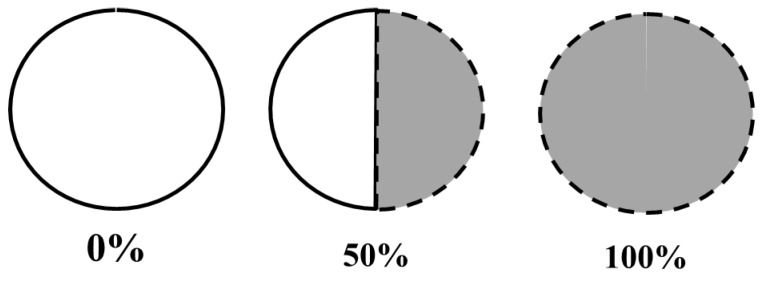
Generalized diagram of clipping by the tuft diameter area division method. Note: the grey part represents the clipping part.

**Figure 2 ijerph-16-01663-f002:**
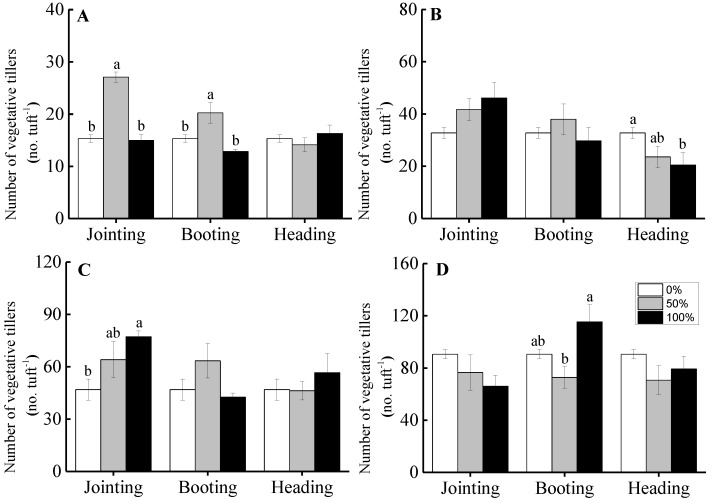
The effect of clipping intensity on the number of vegetative tillers of *H. brevisubulatum* of different grades when clipped at the jointing, booting, and heading periods. **A**, **B**, **C**, and **D** represent the grades (1, 2, 3, and 4, respectively) of the tuft diameters of *H. brevisubulatum*. Different lowercase letters represent significant differences among different clipping intensities (*p* < 0.05).

**Figure 3 ijerph-16-01663-f003:**
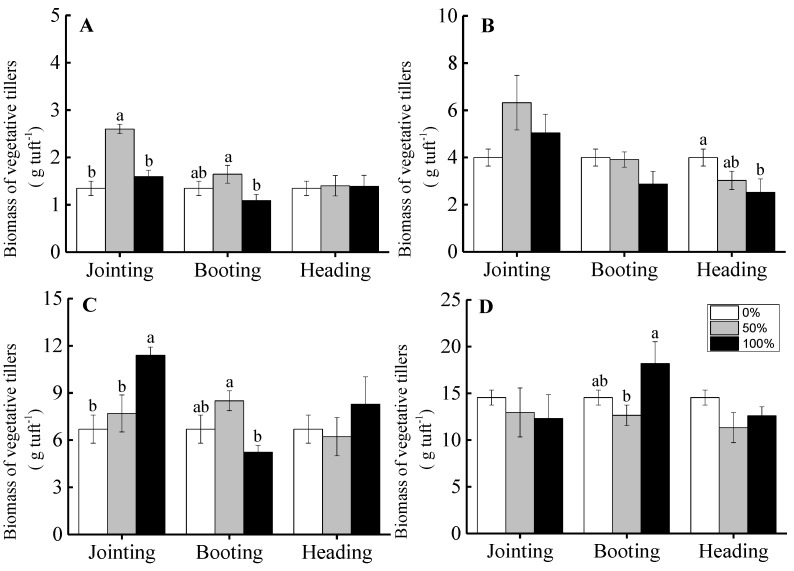
The effects of clipping intensity on the biomass of vegetative tillers of *H. brevisubulatum* of different grades when clipped at the jointing, booting, and heading periods. **A**, **B**, **C**, and **D** represent the grades (1, 2, 3, and 4, respectively) of the tuft diameters of *H. brevisubulatum*. Different lowercase letters represent significant differences among different clipping intensities (*p* < 0.05).

**Figure 4 ijerph-16-01663-f004:**
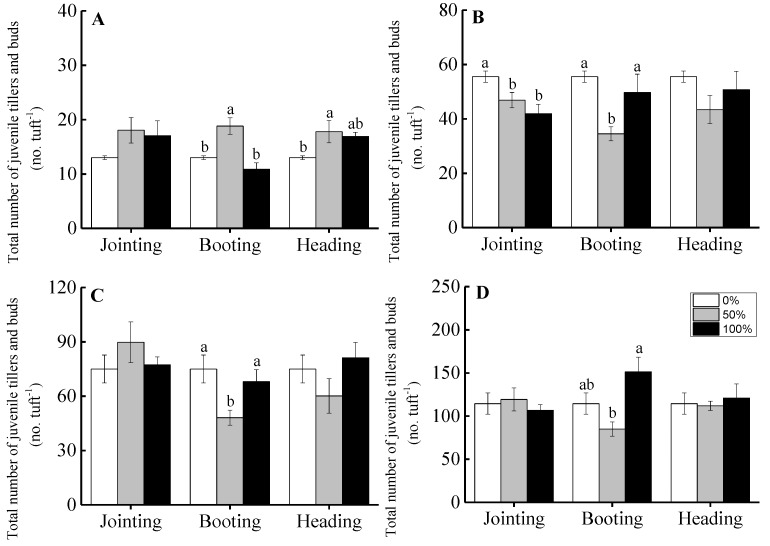
The effects of clipping intensity on total number of juvenile tillers and buds of *H. brevisubulatum* at different grades when clipped at the jointing, booting, and heading periods. **A**, **B**, **C**, and **D** represent the grades (1, 2, 3, and 4, respectively) of the tuft diameters of *H. brevisubulatum*. Different lowercase letters represent significant differences among different clipping intensities (*p* < 0.05).

**Figure 5 ijerph-16-01663-f005:**
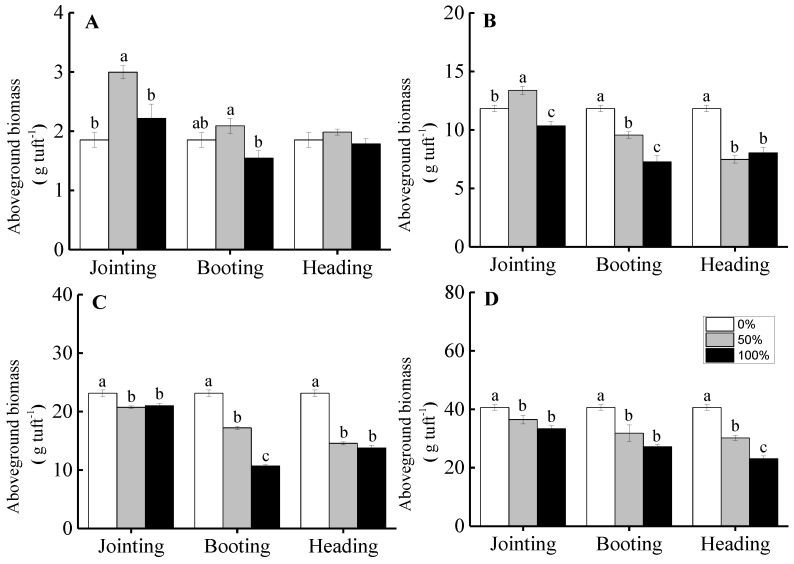
The effect of clipping intensity on the aboveground biomass of *H. brevisubulatum* at different grades when clipping at the jointing, booting, and heading periods. **A**, **B**, **C**, and **D** represent the grades (1, 2, 3, and 4, respectively) of the tuft diameters of *H. brevisubulatum*. Different lowercase letters represent significant differences among different clipping intensities (*p* < 0.05).

**Figure 6 ijerph-16-01663-f006:**
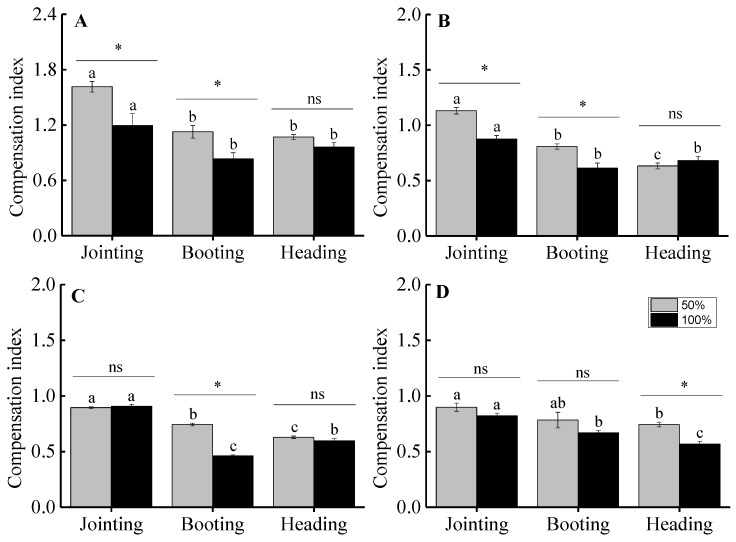
The effects of clipping intensity and phenological period on the compensation index of *H. brevisubulatum* at different grades. **A**, **B**, **C**, and **D** represent the grades (1, 2, 3, and 4, respectively) of the tuft diameters of *H. brevisubulatum*. Different lowercase letters represent significant differences among different phenological periods (*p* < 0.05). *, ns represent that there was a significant difference (*p* < 0.05) and no significant difference (*p* > 0.05) between 50% and 100% clipping intensities, respectively.

**Figure 7 ijerph-16-01663-f007:**
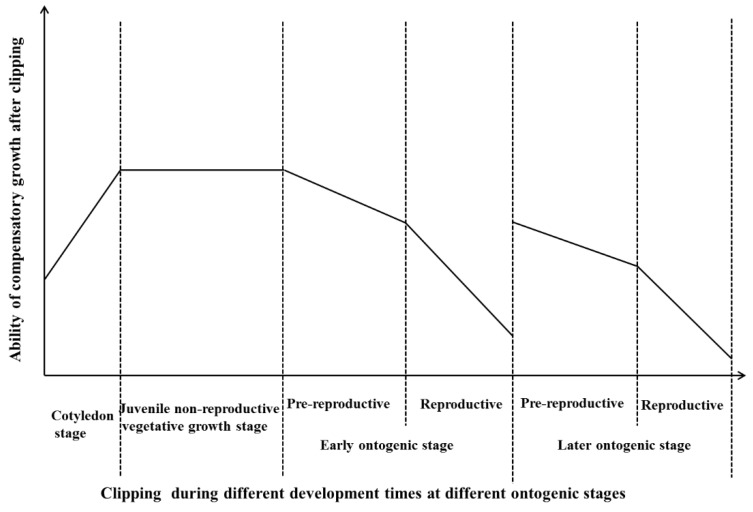
Generalized response diagram of the ability of compensatory growth after clipping during different development times at different ontogenic stages for perennial herbs (synthesized and referred from Boege and Marquis 2005, Hanley and Fegan 2007, Tito et al., 2016, and from this research).

**Table 1 ijerph-16-01663-t001:** Summary of three-way ANOVA analyses for the effects of ontogenic stages (O), phenological periods (P), and clipping intensities (I) on the characteristics of *H. brevisubulatum* (F-values).

Indexes	O	P	I	O × P	O × I	P × I	O × P × I
Number of vegetative tillers	193.04 **	4.08 *	0.38 ns	2.90 *	3.41 **	1.26 ns	3.18 **
Biomass of vegetative tillers (g)	235.34 **	3.16 *	0.43 ns	2.14 ns	2.13 ns	0.90 ns	2.33 **
Total number of juvenile tillers and buds	275.55 **	0.99 ns	3.99 *	1.02 ns	2.08 ns	4.46 **	1.96 *
Aboveground biomass (g)	3195.62 **	71.36 **	209.50 **	8.61 **	46.96 **	19.94 **	5.01 **
Compensation index	115.40 **	114.47 **	71.60 **	3.72 **	4.86 **	7.36 **	4.51 **

ns, *, and ** represent that there was no significant difference (*p* > 0.05), a significant difference (*p* < 0.05), and an extremely significant difference (*p* < 0.01) of different factors and their interaction on characteristics of *H. brevisubulatum*.

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
