# Peer review of "Effects of Simulated Herbivory on the Vegetative Reproduction and Compensatory Growth of Hordeum brevisubulatum at Different Ontogenic Stages"

_ijerph, 2019, doi:10.3390/ijerph16091663_

Round 1

Reviewer 1 Report

The ms is interesting, although only a small number of variables were evaluated. On the other hand, there are a number of areas in the ms which require improvement and clarification. The English should be revised in order to make the manuscript more readable and fluent.Thus, I recommend that the manuscript, in the present form, must be accepted after substantial revision.

Specific points:

The abstract section needs revision. Authors should rephrase the sentences between lines 13-15, 16-19 and 26-27.

Authors should rephrase the sentences in lines 40-41, 67, 98, 112-113, 163, 168-169, 181-182, 210-211, 214-215, 216, 245-246, 250, 294-295, 307, 319-320.

Three treatments (line 112) is really a correct expression in experimental design? Or three factors?

Only 5 plants in each treatment were selected and measured. Why?

The two sentences in lines 178-179 should be included in Figure 2.

Authors used overcompensation and over-compensation along the manuscript. They should standardize the word through the ms.

Author Response

Response to Reviewer 1 Comments

We thank you for the opportunity to major revise our manuscript (ijerph-492714) entitled " Effect of simulated herbivory on vegetative reproduction and  compensatory growth of Hordeum brevisubulatum at different ontogenic stage " to International Journal of Environmental Research and Public Health. We have made changes in response to the comments of the reviewers. The English language of our manscript was edited for proper English language, grammar, punctuation, spelling, and overall style by one or more of the highly qualified native English speaking editors at American Journal Experts.

Point 1: The ms is interesting, although only a small number of variables were evaluated. On the other hand, there are a number of areas in the ms which require improvement and clarification. The English should be revised in order to make the manuscript more readable and fluent.Thus, I recommend that the manuscript, in the present form, must be accepted after substantial revision.

Response 1: The English language of our manuscript has been revised by a native English speaker. Please see the "Track Changes" for detailed information in the revised manuscript.

Point 2: The abstract section needs revision. Authors should rephrase the sentences between lines 13-15, 16-19 and 26-27.

Response 2: Revised and rephased.

Point 3: Authors should rephrase the sentences in lines 40-41, 67, 98, 112-113, 163, 168-169, 181-182, 210-211, 214-215, 216, 245-246, 250, 294-295, 307, 319-320.

Response 3: Revised and rephased.

Point 4: Three treatments (line 112) is really a correct expression in experimental design? Or three factors?

Response 4: Three treatment was revised into three factors. Thanks for the comments, factor is more appropriate here than the treatment (see line 116).

Point 5: Only 5 plants in each treatment were selected and measured. Why?

Response 5: First, because all the 5 replicates were quite similar in size, enough for the statistical demand. Second, more replicates mean occupying more space, and more difficult to control the homogenous environmental conditions.

Point 6: The two sentences in lines 178-179 should be included in Figure 2.

Response 6: Revised.

Point 7: Authors used overcompensation and over-compensation along the manuscript. They should standardize the word through the ms.

Response 7: All were standardized as ‘over-compensation’.

Reviewer 2 Report

The study by Yuan et al was an attempt to clear up past inconsistencies in the results of past studies investigating the effect of clipping experiments on compensatory growth. This study shows clearly what factors need to be considered when making any conclusions of these types of experiments. The paper is mostly logically presented with clear aims and objectives. However, there are several problems that need to be corrected first before the manuscript can be published. Most notable is the poor quality of the English. I strongly recommend that the authors have a native English speaker edit the revised manuscript before re-submitting it. While I could mostly understand what the authors were trying to say, there were times when it was quite difficult, especially in the Discussion.

The introduction is a thorough review of the literature dealing with this topic. The aims are clearly presented. However, question 3 seems to me to be redundant, as it is quite similar to the general objective. I recommend removing question 3.

The methods seem to be quite adequate as are the statistical analyses. Again, the language needs to be corrected to make it more readable.

The authors have separated the Results into four sub-sections. I do not see the need for this. The first two sub-sections dealing with vegetative tillers should be combined. This sub-section could also include the information about juvenile tillers and buds. The sub-section about compensatory growth is okay. The writing is quite wordy and awkward. Re-write this section in a more concise manner. Again, editing by a native English speaker should help with this. Also, Table 2 seems quite complex as I found it difficult to understand what the different letters stood for. Also, some of this information is also shown in Figure 5. It may be possible to put the information in Table 2 into graph form, as with the other data, and if need be have a smaller accompanying table with the ANOVA results if they cannot be put into the graph.

I found the Discussion to be the worst written section of the paper as the writing is very awkward and wordy with several redundant statements. Again, try to write in a more concise manner. I also do not see the need for the sub-sections. If anything, the last two sub-sections dealing with compensatory growth should be combined. In addition, the authors re-state much of their Results but have less information about how their results fit into the wider body of literature dealing with compensatory growth. Some of that information is in the Discussion, but it should be expanded. Lastly, the authors mention (line 267) that the number of juvenile tillers and buds were the main components of vegetative reproduction. It would be good if the authors could include some data concerning this in the Results section, perhaps showing these proportions as a graph or table.

Finally, citations 12 and 46 are the same (Dalgleish et al. 2009. Plant Ecology).  Remove the latter one and change the citation numbers in the text. Make sure to proofread the revised manuscript before re-submitting.

Overall, this can be a good paper. All the authors need to do is to improve their English, again having the revised manuscript controlled and edited by a knowledgeable native English speaker, and make those other small changes to the Results and Discussion sections.

Author Response

Response to Reviewer 2 Comments

We thank you for the opportunity to major revise our manuscript (ijerph-492714) entitled " Effect of simulated herbivory on vegetative reproduction and  compensatory growth of Hordeum brevisubulatum at different ontogenic stage " to International Journal of Environmental Research and Public Health. We have made changes in response to the comments of the reviewers. The English language of our manscript was edited for proper English language, grammar, punctuation, spelling, and overall style by one or more of the highly qualified native English speaking editors at American Journal Experts.

Point 1: The study by Yuan et al was an attempt to clear up past inconsistencies in the results of past studies investigating the effect of clipping experiments on compensatory growth. This study shows clearly what factors need to be considered when making any conclusions of these types of experiments. The paper is mostly logically presented with clear aims and objectives. However, there are several problems that need to be corrected first before the manuscript can be published. Most notable is the poor quality of the English. I strongly recommend that the authors have a native English speaker edit the revised manuscript before re-submitting it. While I could mostly understand what the authors were trying to say, there were times when it was quite difficult, especially in the Discussion.

Response 1: The English language of our manuscript has been revised by a native English speaker.

Point 2: The introduction is a thorough review of the literature dealing with this topic. The aims are clearly presented. However, question 3 seems to me to be redundant, as it is quite similar to the general objective. I recommend removing question 3.

Response 2: The question 3 was removed.

Point 3: The methods seem to be quite adequate as are the statistical analyses. Again, the language needs to be corrected to make it more readable.

Response 3: This manuscript has been revised by a native English speaker.

Point 4: The authors have separated the Results into four sub-sections. I do not see the need for this. The first two sub-sections dealing with vegetative tillers should be combined. This sub-section could also include the information about juvenile tillers and buds. The sub-section about compensatory growth is okay. The writing is quite wordy and awkward. Re-write this section in a more concise manner. Again, editing by a native English speaker should help with this. Also, Table 2 seems quite complex as I found it difficult to understand what the different letters stood for. Also, some of this information is also shown in Figure 5. It may be possible to put the information in Table 2 into graph form, as with the other data, and if need be have a smaller accompanying table with the ANOVA results if they cannot be put into the graph.

Response 4: The first two sub-sections of Results have been combined in one section: 3.1 Vegetative tillers.

Table 2 has been changed into Figure 6. We added the results of over-compensation, complete compensation and under-compensation by the single-sample t-test into sub-section ‘3.3 Aboveground biomass and compensation index of Results’.

Point 5: I found the Discussion to be the worst written section of the paper as the writing is very awkward and wordy with several redundant statements. Again, try to write in a more concise manner. I also do not see the need for the sub-sections. If anything, the last two sub-sections dealing with compensatory growth should be combined. In addition, the authors re-state much of their Results but have less information about how their results fit into the wider body of literature dealing with compensatory growth. Some of that information is in the Discussion, but it should be expanded. Lastly, the authors mention (line 267) that the number of juvenile tillers and buds were the main components of vegetative reproduction. It would be good if the authors could include some data concerning this in the Results section, perhaps showing these proportions as a graph or table.

Response 5: This manuscript has been revised by a native English speaker.

Response: The last two sub-sections dealing with compensatory growth have been combined (see the sub-section 4.2 The effect of clipping on the compensation growth of plant).

Some information about our results fitting into the wider body of literature dealing with compensatory growth in the discussion was added and expanded (see the part of discussion, line 305-313, line 325-331, line 342-347, line 456-459, line 395-405).

We added the data of the proportions of the number of juvenile tillers and buds in the total number of vegetative tillers, juvenile tillers and buds in line 297-301 in the discussion. And the proportions were more than 50% in most cases. The juvenile tillers and buds were the main components of the potential parts of vegetative reproduction. The juvenile tillers and buds develop into tillers when the conditions are suitable, thus further improving the compensatory growth ability in the future.

Point 6: Finally, citations 12 and 46 are the same (Dalgleish et al. 2009. Plant Ecology).  Remove the latter one and change the citation numbers in the text. Make sure to proofread the revised manuscript before re-submitting.

Response 6: The citations 46 was removed and the citation numbers in the text was changed (see the cite number in discussion and the reference).

Point 7: Overall, this can be a good paper. All the authors need to do is to improve their English, again having the revised manuscript controlled and edited by a knowledgeable native English speaker, and make those other small changes to the Results and Discussion sections.

Response 7: Thanks for this positive comment. This manuscript has been revised by a native English speaker.

Round 2

Reviewer 1 Report

I have read the new version of the manuscript and the authors thoughts on all my previous remarks. 

I believe that the manuscript can be accepted for publication.